

# Comparative efficacy of intraoperative radiotherapy and external boost irradiation in early-stage breast cancer: a systematic review and meta-analysis

Jiaxin Liu[1,*], Xiaowei Shi[1,*], Zhenbo Niu[2] and Cheng Qian[2]

[1] Xiamen Hospital, Fudan University Shanghai Cancer Center, Xiamen, China
[2] Affiliated Cancer Hospital of Harbin Medical University, Harbin, China
[*] These authors contributed equally to this work.

## ABSTRACT

External boost radiotherapy (EBRT) and intraoperative radiotherapy (IORT) are shown to be effective in patients with early-stage breast cancer. However, the difference between IORT and EBRT for patients' prognosis remains to be elucidated. The purpose of this meta-analysis is to investigate differences in local recurrence (LR), distant metastases, disease free survival (DFS), and overall survival (OS) between these two therapies. We searched the Cochrane Library, PubMed, Web of Science and Embase, from inception to Jan 10th, 2022. We used The Cochrane risk-of-bias assessment tool to assess the risk of bias of the included studies, and the STATA15.0 tool was used for the meta-analyses. Eight studies were ultimately included. Meta-analysis demonstrated that there was an inconsistent finding in the long-term risk of LR between the two radiotherapies, and there was no significant difference in short-term risk of LR, the metastasis rate, DFS, and OS IORT would be more convenient, less time-consuming, less costly, and more effective at reducing side effects and toxicity. However, these benefits must be balanced against the potential for increased risk of LR in the long term.

Corresponding author
Cheng Qian, 1179525498@qq.com

## INTRODUCTION

Breast radiotherapy is an important adjuvant therapy for patients with early-stage breast cancer, which can greatly improve their prognosis, reduce the risk of recurrence, and increase their survival (*Van Seijen et al., 2021*). Intraoperative radiotherapy (IORT) and external boost radiotherapy (EBRT) are demonstrated to be effective for patients with early-stage breast cancer.The main difference between IORT and EBRT is the therapy duration. IORT is typically performed during breast-conserving surgery and delivers a single dose of radiation to the edge of the tumour bed, while EBRT is performed after the surgery (*Hashemi et al., 2020*). IORT is a special technique for accelerated partial breast irradiation (APBI), which refers to localised irradiation focusing on the tumour
bed (*Veronesi et al., 2001*; *Liljegren et al., 1999*). It includes intraoperative electron radiation therapy (IOERT) and intraoperative X-ray radiation therapy (IOXRT). IOXRT with 20 GY for breast cancers cases and penetration of 0/5 cm that can be defined as boost or radical dose, base on the clinical status, IOERT with radiation of electron in two different doses on boost (11–12 gy) and radical (20–21 Gy) for breast cancer cases (*Beddok et al., 2022*). EBRT includes partial breast external beam radiotherapy and whole breast external beam radiotherapy. Over the past 10 years, a multiple randomised controlled trials have been conducted to evaluate the efficacy of intraoperative IORT and postoperative EBRT in reducing LR, preventing distant metastasis and prolonging DFS and OS in early breast cancer patients (*Vaidya et al., 2010*; *Veronesi et al., 2013*). However, due to the diversity of demographics, histopathology, and systemic treatment modalities in different clinical trials, the comparative efficacy of these two therapies remains controversial (*Keshtgar et al., 2013*; *Huo et al., 2016*). Therefore, we conducted a systematic review and meta-analysis of the efficacy of IORT and EBRT in early breast cancer treatment with a view to providing evidence-based support for clinical decisions.

## METHODS

From the beginning, until January 10, 2022, we searched PubMed, Embase, Cochrane Library and Science for RCT comparing the efficacy of IORT and EBRT in the treatment of early breast cancer. Search was designed using medical-themed headings (MeSH) and freewords.

A reference list of retrieved studies was also searched for possible eligible studies. This meta-analysis is in strict compliance with the System Evaluation and Meta-Analyses Preference Reporting Project (PRISMA). Data were collected as previously described in *Wang, MacInnis & Li (2023)*.

### Inclusion and exclusion criteria

People diagnosed with early breast cancer.

Combination of IORT or EBRT as intervention or control for breast preservation surgery.

Outcome measures including LR, distant metastasis, DFS or OS.

### Exclusion criteria

Non-RCT design (literature review, case report, meeting summary, observation study, etc.).

Participants of less than 10.

Inappropriate outcome measures.

### Study selection and data extraction

All retrieved articles were imported into EndnoteX9, and duplicates were removed. The articles were initially screened *via* browsing titles and abstracts, and the full-texts of potential eligible studies were downloaded and read for further screening. We extracted the following data from the included studies using a pre-designed form: name of the first

author, publication date, patients' nationality, sample size, mean age, tumor stage, tumor size, lymph node metastasis, follow-up duration, and outcome measures. When different follow-up periods of one single RCT reported by multiple articles, we summarize the most comprehensive outcome indicators and different follow-up times. Study selection and data extraction were conducted by two reviewers independently (LJX and SXW), cross-checked by each other. Any disagreement was settled *via* consulting a third reviewer (QC).

## Quality assessment

Two independent reviewers assessed the risk of bias included in the study using the Cochrane risk-of-bias Assessment Tool. Two researchers (LJX and SXW) then cross-checked their work. If there is any objection, it shall be resolved through consultation with the third examiner. The Cochrane risk-of-bias assessment tool contains the following six domains: Selection bias (stochastic sequence generation and allocation concealment), performance bias (participant and population bias), detection bias (outcome assessment bias), attrition bias (incomplete outcome data), reporting bias (selective reporting), and other bias. Each can be classified as "high", "low" or "unclear". In addition, a the NOS scale (Newcastle-Ottawa Scale) was applied to assess the quality of the study, using a propensity score matching subgroup (*Stang, 2010*), which contains participants' selection (four projects), comparability (one item) and outcome evaluation (three projects), for a total score of 9. The test scores were 7–9 for high quality.

## Data analysis

Stata 15.0 (StataCorp LLC, College Station, TX) software for metaanalysis. Risk ratio (RR) acts in combination with confidence interval (95% CI) for LR and distant metastasis. Hazard ratio (HR) of 95% CI was used to aggregate DFS and OS. Heterogeneity test were performed using Cochrane Q assay and Higgins I2 statistic. I2 showed no significant, moderate, significant and significant heterogeneity in the range of 0–25%, 26%–50%%, 51%–75% and 76%–100%, The statistical methods of meta-analysis include fixed effect model and random effect model. The fixed effect model assumes that each independent study comes from the same population, and the variability between different studies is very small.Random effect model means that each study comes from different populations, and each study has great variability.

# RESULTS

## Study selection

Detailed study selection process is shown in Fig. 1. A total of 1013 relevant articles were retrieved (PubMed = 109, Embase = 292, Cochrane = 126, Web of Science = 486). After removing duplicates ($n = 332$), 607 irrelevant articles were excluded, and full-texts of the remaining 74 articles were read. Finally, eight studies were included in this meta-analysis. It should be noted that although *Veronesi et al. (2013)* and *Orecchia et al. (2021)* were both based on ELIOT clinical trial, *Veronesi et al. (2013)* reported DFS, which was not reported by *Orecchia et al. (2021)*. Similarly, based on TARGIT-A clinical trial, *Yasser et al. (2019)* reported exclusive outcome indicators of distant metastasis, which was not reported in the

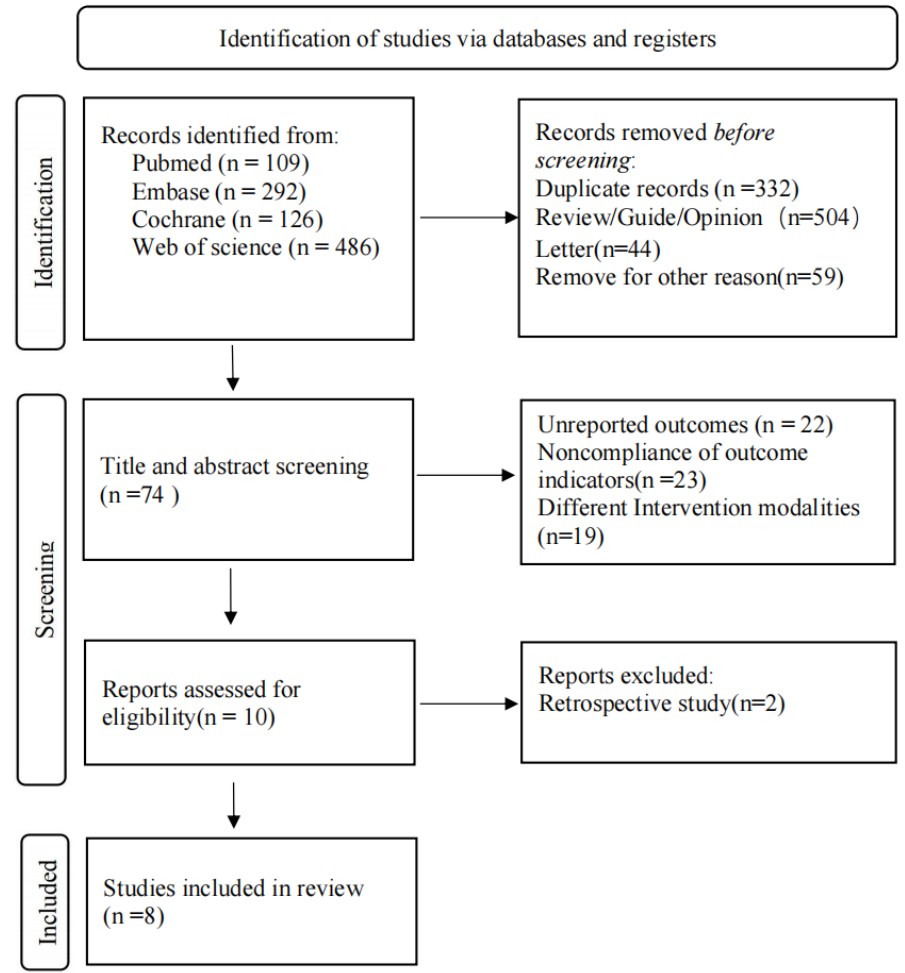

**Figure 1   Flow diagram of the study selection process.**

study by *Vaidya et al. (2016)*. Therefore, we still included these two studies, even though their sample sources were partially duplicated. We did not calculate the total sample size due to repeated publication, but we conducted subgroup analyses of the studies.

## Characteristics of included studies

Detailed characteristics of included studies are shown in Table 1. Among the included studies, one study was retrospective-design and the rest of the 7 were all RCT. Most of the studies were multi-regional and multi-centered. The study by *Vaidya et al. (2016)*, involved 11 countries with 33 centers including the United Kingdom (714), Australia (394), Italy (476), Germany (734), the United States (266), Poland (42), Denmark (514), Canada (24), Switzerland (98), Norway (111), and France(78). The study population of *Mi et al. (2020)* was Chinese. The included studies were published between 2013 and 2021 (median: 2020). Most participants were younger than 60, with a tumor grade of G2 and a T1 size but no lymph node metastasis.

Peerj

**Table 1 Detailed characteristics of included studies.**

| Author | Year | Country | Number of cases | | Age | | Tumor staging | | Tumor size | | Lymph node metastasis | |
|---|---|---|---|---|---|---|---|---|---|---|---|---|
| | | | IORT | EBRT | IORT | EBRT | IORT | EBRT | IORT | EBRT | IORT | EBRT |
| Vaidya JS | 2016 | 33 centers in 11 countries | 1721 | 1730 | <60:677 ≥60:781 | <60:670 ≥60:807 | G1 538 G2 757 G3 232 | G1 558 G2 720 G3 227 | T1:1362 T2:190 | T1:1323 T2:207 | None: 1348 Visible:194 | None: 1343 Visible:178 |
| | 2020 | | 581 | 572 | <60:196 ≥60:385 | <60:194 ≥60:284 | G1 305 G2 204 G3 31 | G1 339 G2 159 G3 33 | T1:543 T2:33 | T1:533 T2:27 | None: 536 Visible:30 | None: 533 Visible:19 |
| | 2021 | | 1140 | 1158 | <60:953 ≥60:1345 | <60:474 ≥60:684 | G1 275 G2 621 G3 226 | G1 286 G2 615 G3 217 | T1:949 T2:176 | T1:927 T2:190 | None: 931 Visible:185 | None: 946 Visible:172 |
| YA Madyan | 2019 | Germany | 90 | 90 | Median:64 | Median:65 | G1 18 G2 56 G3 16 | G1 19 G2 55 G3 16 | T1:79 T2:11 | T1:76 T2:14 | None: 74 Visible:16 | None: 71 Visible:19 |
| Y Mi | 2020 | China | 82 | 199 | <60:41 ≥60:23 | <60:42 ≥60:22 | G1 5 G2 50 G3 9 | G1 9 G2 42 G3 13 | T1:47 T2:17 | T1:51 T2:13 | None: 57 Visible:7 | None: 55 Visible:9 |
| U Veronesi | 2013 | European Institute of Oncology | 651 | 654 | <60:330 ≥60:123 | <60:310 ≥60:344 | G1 196 G2 305 G3 129 | G1 160 G2 328 G3 145 | T1:562 T2:83 | T1:554 T2:103 | None: 478 Visible:169 | None: 471 Visible:176 |
| R Orecchia | 2021 | | 651 | 654 | <60:310 ≥60:344 | <60:330 ≥60:321 | G1 160 G2 328 G3 145 | G1 196 G2 305 G3 129 | T1:544 T2:103 | T1:562 T2:83 | None: 478 Visible:169 | None: 471 Visible:176 |
| A Ciabattoni | 2021 | Italy | 125 | 110 | Average 56.3 | Average 56.2 | G1-2:81 G3: 44 | G1-2:73 G3:27 | T1:96 T2:28 | T1:79 T2:22 | None: 82 Visible:43 | None: 68 Visible:42 |

In addition, Table 2 provided the outcome indicator information about LR, distant metastasis, DFS, and OS for both the two groups of patients in all studies.

## Quality assessment

The included studies were mainly RCT, with open-label design, which exerted no substantial impact on the assessment of the results. The study by *Mi et al. (2020)* were retrospective, which used propensity score matching method. The retrospective studies were evaluated by the NOS scale and scored for 8 (Table 3).

# META-ANALYSIS RESULTS

## Local recurrence

There were four studies (*Vaidya et al., 2016*; *Mi et al., 2020*; *Ciabattoni et al., 2021*; *Orecchia et al., 2021*) that reported LR. Random-effect model was used due to significant heterogeneity among the studies ($I^2 =89.3\%$). Meta-analysis showed that there was no significant difference in the short-term risk of LR between IORT group and EBRT group [RR $=1.90$, 95%CI (0.73, 4.96), $P = 0.190>0.05$], which was inconsistent with the long-term risk of LR between the two groups [10-year RR $=2.78$, 95%CI(0.41, 18.80)P $=0.295.>0.05$] [12-year RR $=1.05$, 95%CI(0.85, 1.31)P $=0.647.>0.05$] [15-year RR $=4.52$, 95%CI (2.74, 7.45)P $=0.000<0.05$] (Fig. 2A). Sensitivity analysis showed that after removing each study one by one, the results did not reverse, indicating the robustness of the results (Fig. 2B). The funnel plot visually illustrated the publication bias of each study, and the Egger's test showed no publication bias (Fig. 2C).

## Distant metastasis

There were four studies (*Mi et al., 2020*; *Ciabattoni et al., 2021*; *Orecchia et al., 2021*; *Yasser et al., 2019*) that reported the incidence of distant metastasis. No heterogeneity was observed among the studies ($I^2 =0.0\%$), and fixed-effect model was applied. Meta-analysis showed that there was no significant difference in the incidence of distant metastasis between the two groups [5-year RR $=0.93$, 95%CI (0.58, 1.50), $P = 0.778>0.05$] [10-year RR $=0.80$, 95%CI (0.54, 1.19), $P = 0.278>0.05$] (Fig. 3A). This indicated that IORT and EBRT had similar risk for distant metastasis. Sensitivity analysis showed that the removal of each study one by one did not reverse the results, indicating its robustness (Fig. 3B). The funnel plot visually illustrated the publication bias of each study, and the Egger's test showed no publication bias (Fig. 3C).

## Disease-free survival

There were five studies (*Veronesi et al., 2013*; *Vaidya et al., 2016*; *Mi et al., 2020*; *Ciabattoni et al., 2021*; *Vaidya et al., 2020a*) that reported DFS. No heterogeneity was observed among the studies ($I^2 =0.0\%$), and fixed-effect model was applied. Meta-analysis showed that there was no significantly statistical difference in the DFS between the two groups [5-year RR $=1.09$, 95%CI (0.84, 1.41), $P = 0.506>0.05$] [10-year RR $=0.97$, 95%CI (0.73, 1.28), $P = 0.810>0.05$] (Fig. 4A). This indicated that IORT and EBRT had similar DFS. Sensitivity analysis showed that the removal of each study one by one did not reverse the results,

**Table 2 Characteristics of included studies.**

| Author | Year | Follow Time | LR | | | | | | Metastasis | | | | | |
|---|---|---|---|---|---|---|---|---|---|---|---|---|---|---|
| | | | IORT | | | EBRT | | | IORT | | | EBRT | | |
| | | | Event | N-Event | Total | Event | N-Event | Total | Event | N-Event | Total | Event | N-Event | Total |
| Vaidya, JS | 2016 | 5 | 23 | 1698 | 1721 | 11 | 1719 | 1730 | / | / | / | / | / | / |
| | 2020 | 10 | / | / | / | / | / | / | / | / | / | / | / | / |
| | 2021 | 12 | 144 | 996 | 1140 | 139 | 1019 | 1158 | / | / | / | / | / | / |
| YA Madyan | 2019 | 5 | / | / | / | / | / | / | 3 | 87 | 90 | 2 | 88 | 90 |
| Y Mi | 2020 | 5 | 3 | 79 | 82 | 6 | 193 | 199 | 0 | 82 | 82 | 5 | 194 | 199 |
| U Veronesi | 2013 | 5 | / | / | / | / | / | / | / | / | / | / | / | / |
| R Orecchia | 2021 | 5 | 33 | 618 | 651 | 5 | 649 | 654 | 29 | 622 | 651 | 31 | 623 | 654 |
| | | 10 | 64 | 587 | 651 | 9 | 645 | 654 | 39 | 612 | 651 | 49 | 605 | 654 |
| | | 15 | 81 | 570 | 651 | 18 | 636 | 654 | / | / | / | / | / | / |
| A Ciabattoni | 2021 | 5 | 7 | 118 | 125 | 9 | 101 | 110 | / | / | / | / | / | / |
| | 2021 | 10 | 12 | 113 | 125 | 10 | 100 | 110 | 3 | 122 | 125 | 3 | 107 | 110 |

| Author | Year | Follow time | DFS | | | | | | OS | | | | | |
|---|---|---|---|---|---|---|---|---|---|---|---|---|---|---|
| | | | IORT | | | EBRT | | | IORT | | | EBRT | | |
| | | | Event | N-Event | Total | Event | N-Event | Total | Event | N-Event | Total | Event | N-Event | Total |
| Vaidya, JS | 2016 | 5 | 78 | 1643 | 1721 | 69 | 1661 | 1730 | 37 | 1684 | 1721 | 51 | 1679 | 1730 |
| | 2020 | 10 | 62 | 510 | 572 | 62 | 519 | 581 | 56 | 525 | 581 | 62 | 510 | 572 |
| | 2021 | 12 | / | / | / | / | / | / | 110 | 1030 | 1140 | 131 | 1027 | 1158 |
| YA Madyan | 2019 | 5 | / | / | / | / | / | / | / | / | / | / | / | / |
| Y Mi | 2020 | 5 | 3 | 79 | 82 | 12 | 187 | 199 | 2 | 80 | 82 | 4 | 195 | 199 |
| U Veronesi | 2013 | 5 | 23 | 628 | 651 | 20 | 634 | 654 | / | / | / | / | / | / |
| R Orecchia | 2021 | 5 | / | / | / | / | / | / | 21 | 630 | 651 | 21 | 633 | 654 |
| | | 10 | / | / | / | / | / | / | 61 | 590 | 651 | 61 | 593 | 654 |
| | | 15 | / | / | / | / | / | / | 108 | 543 | 651 | 115 | 539 | 654 |
| A Ciabattoni | 2021 | 5 | 11 | 114 | 125 | 10 | 100 | 110 | 7 | 118 | 125 | 1 | 109 | 110 |
| | 2021 | 10 | 20 | 105 | 125 | 21 | 89 | 110 | 10 | 115 | 125 | 6 | 104 | 110 |

**Table 3** Quality assessment of included studies.

| Author | Year | v1 | v2 | v3 | v4 | v5 | v6 | v7 |
|--------|------|-----|------|-----|-----|-----|-----|-----|
| Vaidya, JS | 2016 | Low | High | Low | Low | Low | Low | Low |
| Vaidya, JS | 2020 | Low | High | Low | Low | Low | Low | Low |
| Vaidya, JS | 2021 | Low | High | Low | Low | Low | Low | Low |
| YA Madyan | 2019 | Low | High | Low | Low | Low | Low | Low |
| U Veronesi | 2013 | Low | High | Low | Low | Low | Low | Low |
| R Orecchia | 2021 | Low | High | Low | Low | Low | Low | Low |
| A Ciabattoni | 2021 | Low | High | Low | Low | Low | Low | Low |

**Notes.**

v1, Random sequence generation; v2, Allocation concealment; v3, Performance bias; v4, Detection bias (Blinding of outcome assessment); v5, Attrition bias (Incomplete outcome data); v6, Reporting bias (Selective reporting); v7, Other bias.

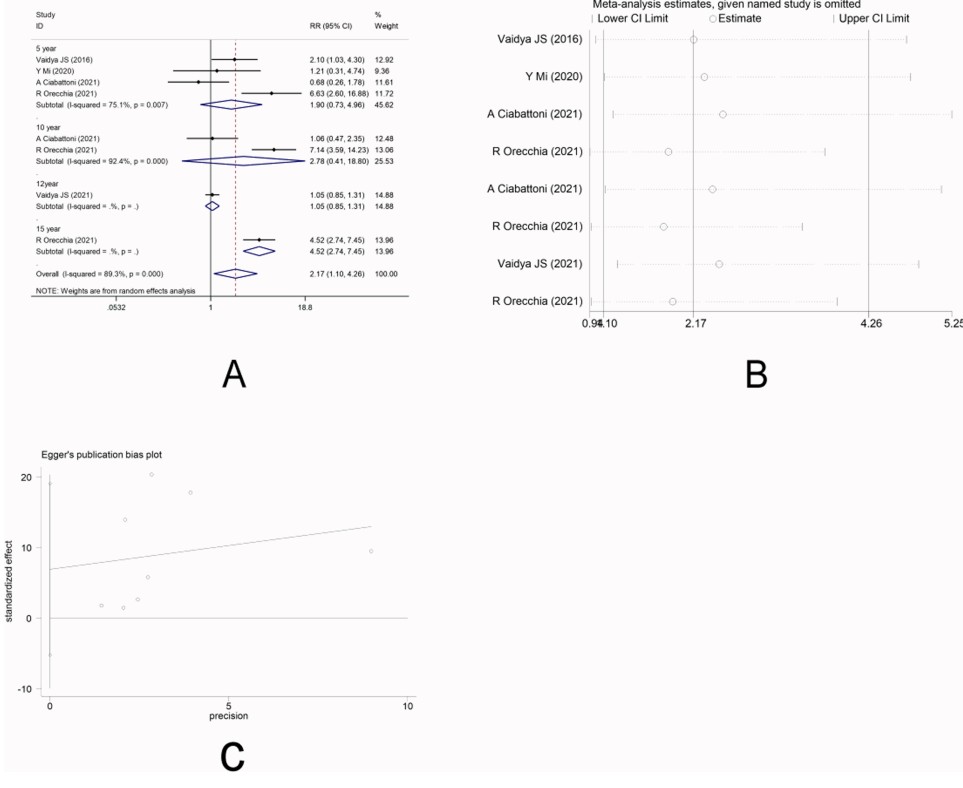

**Figure 2** (A–C) Analysis of local recurrence.

indicating its robustness (Fig. 4B). The funnel plot visually illustrated the publication bias of each study, and the Egger's test showed no publication bias (Fig. 4C).

## Overall survival

There were six studies (*Vaidya et al., 2016*; *Mi et al., 2020*; *Ciabattoni et al., 2021*; *Orecchia et al., 2021*; *Vaidya et al., 2020a*; *Vaidya et al., 2021*) that reported OS. No heterogeneity was observed among the studies ($I^2 = 0.0\%$), and fixed-effect model was applied. Meta-analysis
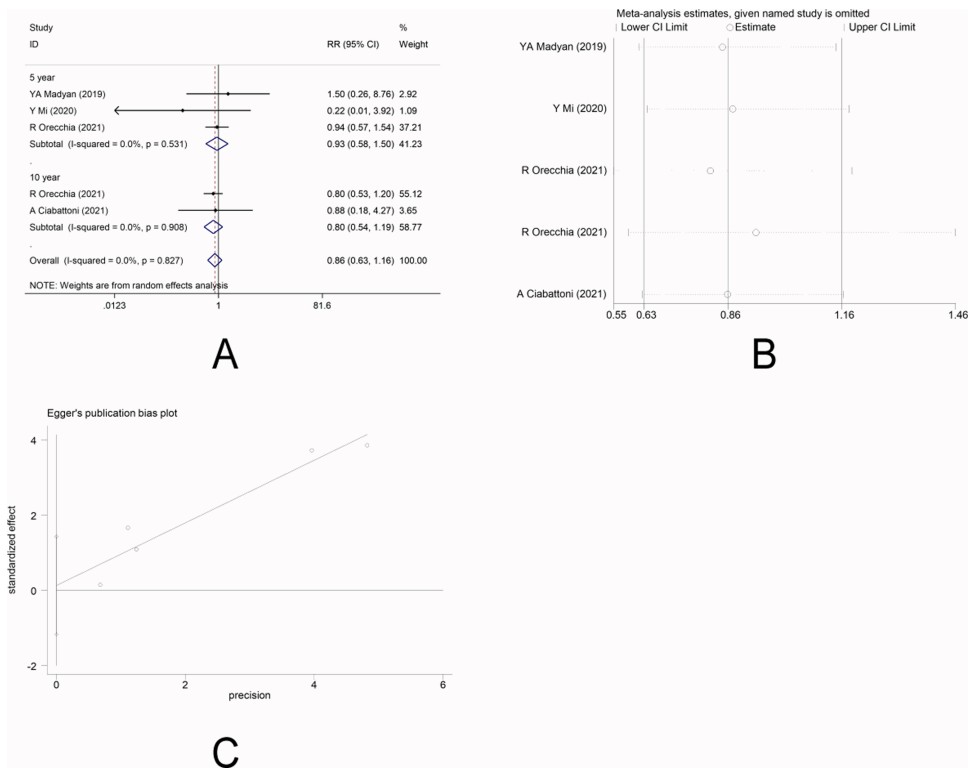

**Figure 3** (A–C) Analysis of distant metastasis.

showed that there was no significant difference in OS between the two radiotherapy methods [5-year RR =0.96, 95%CI (0.58, 1.58), $P = 0.865 > 0.05$] [10-year RR =0.97, 95%CI (0.77, 1.22), $P = 0.797 > 0.05$] [12-year RR =0.85, 95%CI (0.67, 1.08) $P = 0.194 > 0.05$] [15-year RR =0.94, 95%CI (0.74, 1.20), $P = 0.633 > 0.05$] (Fig. 5A). This indicated that IORT and EBRT had similar OS. Sensitivity analysis showed that the removal of each study one by one did not reverse the results, indicating its robustness (Fig. 5B). The funnel plot visually illustrated the publication bias of each study, and the Egger's test showed no publication bias (Fig. 5C).

# DISCUSSION

This meta-analysis indicates that the difference in the efficacy of IORT and EBRT in preventing long-term LR remains elusive in patients with early-stage breast cancer, and both these two methods have no effects on preventing the short term LR (within 5 years). In addition, there were no significant differences in reducing the risk for distant metastasis risk, DFS, and OS ($P > 0.05$) between IORT and EBRT, This result shows that:IORT would be more convenient, time-saving, and cost-effective, and would be more effective in reducing the side effects and toxicity. However, these advantages must be weighed against the possibility of increasing the risk of long-term LR.which is in consistence with the study by *Kolberg et al. (2017)*. They have found no significant difference between IORT and EBRT

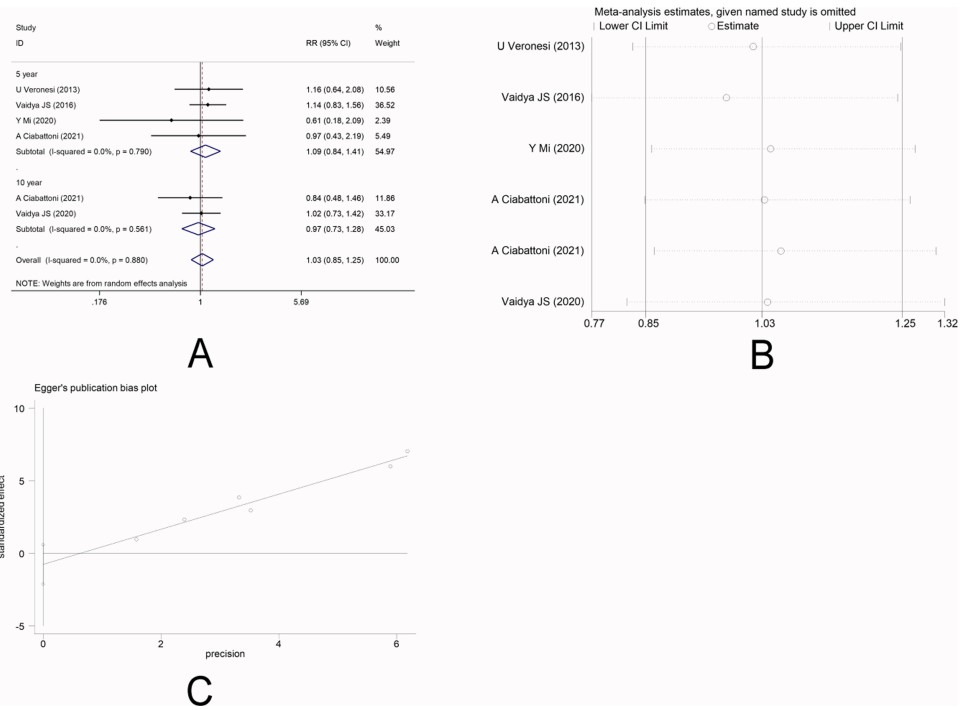

**Figure 4** (A–C) Analysis of the disease-free survival.

in terms of LR, distant relapse or any relapse. IORT might be more effective than EBRT in prolonging LR-free survival and DFS, while the difference is statistically unsignificant.

A study by *Moini et al. (2020)* also found that within 54 months, IORT patients had three (1.2%) LR, while EBRT patients had eight (2.5%) LR ($P = 0.361$) and 12 (4.7%) LR metastasis. the EBRT group 20 (6.2%) ($P = 0.724$). The 5-year DFS (DFS) was 85.1% in the IORT group, compared with 86% in the EBRT group. 50 kV X-ray IORT tumor bed boost were effective in breast preservation treatment, but there was no significant difference compared to EBRT. These results suggest that IORT and EBRT contribute similarly to the OS in patients with early-stage breast cancer, and only part of the patients may face the risk of long-term LR. Is it common in the tumor characteristics in patients with LR? The association between the characteristics of patients receiving IORP and the incidence of LR was assessed by *Veronesi et al. (2013)* and *Hein et al. (1986)* and found that larger tumor size (>2 cm), grade-3 tumor features with more than 4 positive lymph nodes, and triple negative breast cancer were significantly associated with LR. In view of these tumor features with high recurrence risk, would IORT combined with EBRT be beneficial compared with single EBRT? A preliminary study by *Vaidya et al. (2010)* nonrandomized patients who received IORT+EBRT to those who received EBRT only and found significant reductions in non-breast cancer mortality (0/218 *vs* 24/892, $p = 0.012$). A study by *Yasser et al. (2019)* also compared IORT+EBRT with EBRT in survival-improving, and the results showed that the 5-year risk for LR was 0% in IORT+EBRT group, and 1.1% in EBRT group. However, this result might be affected by limited sample size ($n = 90$).

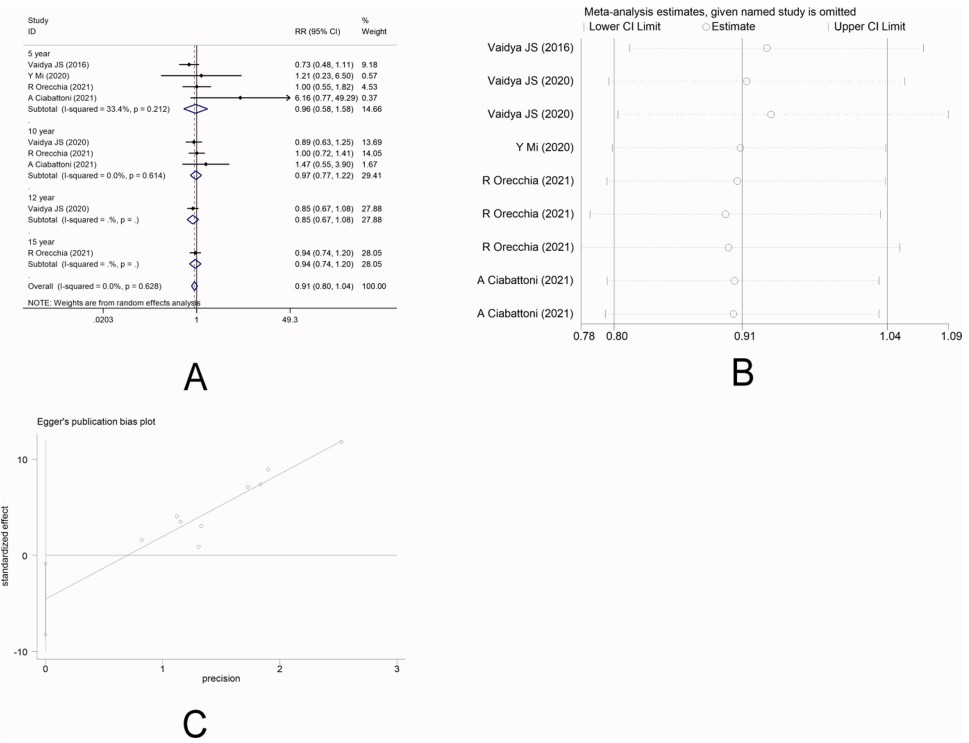

**Figure 5** (A–C) Analysis of the overall survival.

IORT includes IOERT and intraoperative X-ray radiation therapy IOXRT, Four of the RCT we included in the analysis were IOXRT (*Vaidya et al., 2016*; *Vaidya et al., 2020a*; *Vaidya et al., 2021*; *Mi et al., 2020*) and three were IOERT (*Veronesi et al., 2013*; *Orecchia et al., 2021*; *Ciabattoni et al., 2021*). Does different radiotherapy methods, dose and timing affect treatment outcome? In a study by *Hashemi et al. (2021)*, IORT was delivered with both X-ray and electrons, and each group was divided into radical and boost dose subgroups, and the efficacy of this modality with the control group that received WBRT was compared. With a mean follow-up of 34.5 and 40.18 months for the IORT and EBRT groups, respectively, there was a significant difference in DFS between electron boost and X-ray boost groups ($P = 0.037$) and the electron radical group compared with EBRT ($P = 0.025$), but there was no significant difference between other boost and radical groups in DFS and OS.

A study by *Vaidya et al. (2020b)* assessed the effect of IORT and WBI delays on LR and Survival. The 5-year incidence of LR was: delayed TARGIT-IORT *vs* EBRT [23/581(3.96%) *vs* 6/572 (1.05%); difference 2.91%; upper 90% CI *vs* 4.4%]. Long-range follow-up (median [IQR], 9.0 [7.5–10.5] years], LR-free survival (HR, 0.75; 95% CI [0.57–1.003]; $P = 0.52$), mastectomy free survival (HR, 0.88; 95% CI [0.65–1.18]; $P = 0.38$), CI [0.72–1.39]; $P = 0.98$, 0.95%These long-term data show no statistically significant decrease in mastectomy survival, distant DFS, or OS, despite an increase in the number of patients with LR in delayed TARGIT-IORT group. Tumor radiotherapy improves local control

and survival, as well as multiple adverse reactions, including cardiotoxicity and secondary malignancies (*Henson et al., 2013*). *Veronesi et al. (2013)* compared the skin side effects involved in IORT and EBRT and found that the skin side effects in the IORT group were less than those in EBRT group, with erythema ($P < 0.0001$), dryness ($P = 0.04$), pigmentation($P = 0.0004$), and pruritus ($P = 0.002$). *Vaidya et al. (2020a)*, *Vaidya et al. (2021)*, *Sarles et al. (1989)*, and *Preskorn et al. (2022)* demonstrated a similar incidence of complications and severe toxicity in patients receiving IORT and EBRT [severe toxicity: Targit 37/1113 (3.3%) vs.. EBRT 44/1119 (3.9%); $p = 0.44$). The incidence of radiation toxicity was lower in the TARGIT group (6 cases, 0.5%) than in EBRT group (23 cases, 2.1%; $p = 0.002$). Among the complications six months after surgery, the incidence of wound-related complications was generally the same between the two groups, while TARGIT significantly had lower incidence of grade-3 or grade-4 skin complications (4/1720 *vs.* 13/1731; $p = 0.029$). For patients undergoing breast reconstruction, IORT would be more preferable (*Krivorotko et al., 2021*; *Fertsch et al., 2017*) Long-term and continuous external breast irradiation could cause contracture of dilator or prosthesis, asymmetry even deformity of breast morphology and healthy side, so that cause complications such as prosthesis rupture and dilator infection. The use of IORT can effectively avoid this issue. The prosthesis or dilator can be implanted after radiotherapy to avoid radiotherapy radiation. The breast skin will not be changed after radiotherapy. Compared with EBRT, IORT also has certain merits in health economics (*Shah et al., 2014*). If an IORT is used in the right patient instead of an EBRT, it could save healthcare providers between £8 million and £9.1 million annually. This does not include environmental, patient, and social costs (*Vaidya et al., 2017*). Patients receiving IORT would not need to visit the radiation center every day for weeks, and even in 2015, in a modern urban community, New Jersey, patients who lived more than 9.2 miles from a radiation facility had a 36% higher chance of getting a mastectomy than those lived less than 9.2 miles from the facility (*Preskorn et al., 2022*). Therefore, IORT presents to be more convenient, time-saving, and cost-effective for the patients, so that improves their quality of life and reduce the risk for side effects and toxicity. However, these advantages must be weighed against the possibility of increasing the risk of LR.

Our study had the following strengths: Firstly, we included RCTs and propensity-matching scoring subgroup study of high-quality. Secondly, we assessed the short-term (5-year) and long-term (15-year) follow-up outcomes. Thirdly, our meta-analysis involved a large sample size, and the participants covered multiple centers in various regions so that avoided possible racial and social impacts to a large extent. However, several limitations also existed. Although we reflect the results of the long-term follow-up, the study on each index was rarely reported. Our discussion of side effects, toxicity, and cost was also limited to selected articles and did not synthesize a systematic analysis of all relevant studies. These limitations are expected to be resolved in future by more clinical trials and meta-analyses of high quality.

## CONCLUSION

IORT and EBRT have similar short-term LR risk in early breast cancer patients, but their impact on long-term LR remains unclear. There is no significant difference between the two approaches in reducing the risk of distant metastasis in early breast cancer and improving DFS and OS. Using IORT is more practical, less time-consuming, less costly, and more effective at reducing side effects and toxicity. However, these benefits need to be balanced against the potential for long-term increases in LR risk.We have registered the concrete details on Inplasy, DOI: 10.37766/inplasy2023.5.0025 (*Liu et al., 2023*).

### Funding

The authors received no funding for this work.

### Competing Interests

The authors declare there are no competing interests.

### Author Contributions

- Jiaxin Liu conceived and designed the experiments, authored or reviewed drafts of the article, and approved the final draft.
- Xiaowei Shi conceived and designed the experiments, authored or reviewed drafts of the article, and approved the final draft.
- Zhenbo Niu analyzed the data, prepared figures and/or tables, and approved the final draft.
- Cheng Qian performed the experiments, analyzed the data, prepared figures and/or tables, and approved the final draft.

### Data Availability

The raw data is available in the Supplemental Files.

### Supplemental Information

Supplemental information for this article can be found online at http://dx.doi.org/10.7717/peerj.15949#supplemental-information.

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
