# Peer review of "Comparative efficacy of intraoperative radiotherapy and external boost irradiation in early-stage breast cancer: a systematic review and meta-analysis"

_PeerJ, doi:10.7717/peerj.15949_

## Round 0.1 · original submission · Major Revisions

This intriguing study compares the efficacy of IORT and EBRT in the early treatment of breast cancer. Nonetheless, as suggested by our Reviewers, a number of issues need to be addressed to enhance the quality and clarity of this MS. Before the MS can be considered for publication, I recommend that the authors revise it to resolve the following issues and other issues that were raised by our Reviewers.

Here are my comments:
1. It is essential that the title of a paper conveys the main topic and purpose of the study concisely. The title needs improvement. For example, "Comparative Efficacy of Intraoperative Radiotherapy and External Boost Irradiation in Early-Stage Breast Cancer: A Systematic Review and Meta-Analysis".
2. In the section on Data Analysis, the statistical methods are not explicitly described. To assure that the analysis can be replicated, greater specificity is required. The authors should define "stochastic effect models" and explain how they determined when to use them.
3. The Abstract section also requires revision. What useful information does this article's findings provide to clinicians and researchers? The present abstract lacks a concluding statement and summary.
4. Results Interpretation: The authors should provide additional results interpretation and discussion. In addition to stating that there was no significant difference in Disease-Free Survival (DFS) and Overall Survival (OS) between the IORT and EBRT groups, the authors should explain what this means in the context of the current literature and clinical practice.
5. Include actual p-values when reporting results, rather than simply asserting "P>0.05" This would provide greater detail regarding the statistical significance of their findings.
6. Grammatical errors and awkward phrases are scattered in the text. For example, based on the clinical condition? Localised?

Reviewer 1 ·

Basic reporting

The research problem has practical clinical significance, research methods has scientific validity.

Experimental design

Experimental design is basically reasonable.

Validity of the findings

The results and findings are mainly valid and have certain clinical significance.

Additional comments

The article has certain research significance, and can be published after modifying the language problems

Reviewer 2 ·

Basic reporting

no comment

Experimental design

no comment

Validity of the findings

no comment

Additional comments

It is a fine analysis, For patients with early breast cancer, the short-term risk of LR was similar in IORT group and EBRT group, while the long-term risk for recurrence presented inconclusive, and no significant differences observed in distant metastasis, DFS, and OS between the two groups. Authors conducted a systematic review and meta-analysis of the efficacy of IORT and EBRT in early breast cancer treatment with a view to providing evidence-based support for clinical decisions. However, the inclusion considerations of different articles from the same clinical trial need to be explained. noncomplete Figures did not format correctly and are difficult to read. Abstract has several sentences that should be revised/edited for clarity if manuscript is to be resubmitted.
Here I am going to introduce some comment that can help the authors to do the best

1)Line 8-9: How to understand”Meta-analysis showed that there was an inconsistent conclusion in long-term risk of LR between the two radiotherapies and no significant difference in short-term risk”

2)Line 24-25: The pooled results showed that there was a significant difference in local recurrence between the two radiotherapy methods (RR=2.04 [95%CI:.....
Comment: This is true only for 10 years or longer

3)Line 40-41: Breast radiotherapy has become an important adjuvant therapy
Comment: Line 40-41: It should be - Breast radiotherapy is an important adjuvant therapy

4)Line 43-44: Intraoperative radiation therapy (IORT) is a kind of (special technique for) accelerated partial breast irradiation (APBI)
Comment: It should be - IORT is a special technique for accelerated partial breast irradiation (APBI)

5)Line 46: IORT is usually combined with breast-conserving
Comment: It should be - IORT is combined with breast-conserving

6)Line 92-94 :Most of the studies were multi-regional and multi-centered. The study by Vaidya JS et al, involved 11 countries with 33 centers including the United Kingdom (714), Australia (394), Italy (476), Germany (734), the United States (266), Poland (42), Denmark (514), Canada (24), Switzerland (98), Norway (111), and France (78). The study population of the study by Y Mi was Chinese.
Comment: It should be - All patients come from multi-regional and multi-centered including the United Kingdom (714), Australia (394), Italy (476), Germany (734), the United States (266), Poland (42), Denmark (514), Canada (24), Switzerland (98), Norway (111), France (78), China(281)

7)Line 155-160: IORT accounted for 82% and EBRT 66% in patients aged 50-70 years. IORT accounted for 83% and EBRT 88% in tumor size less than 2cm. IORT accounted for 85% and 13% in in the groups of G1-2 and G3 tumor, respectively. EBRT accounted for 84% and 14% in the groups of G1-2 and G3 tumor, respectively. IORT accounted for 78% and EBRT 76% in the group without metastasis into lymph node.
Comment: The term ‘accounted for’ is not appropriate and the sentence should be modified.

Reviewer 3 ·

Basic reporting

no comment

Experimental design

-Table 2 is Incomplete without data of DFS and OS.
-The entry represented by V1-7 shall be indicated in Table 3.

Validity of the findings

There are multiple studies of the same patient population (TARGIT-A and 10 year follow-up) and ELIOT v.1 and follow up. The principles of the meta analysis are therefore not valid as there are duplicate patient cohorts/results analyzed.
(1)The Italian ELIOT RCT has been included twice (U Veronesi 2013 and R Orecchia 2021)
(2)TARGIT RCT has also been included thrice (Vaidya 2013, 2020 and 2021).

Additional comments

This article summarizes and analyzes the differences between intraoperative radiotherapy and external radiotherapy in OS, DFS, LR and metastasis.get the conclusion similar to clinical experience. However, it is recommended that the author make the following modifications:

-In the abstract, please confirm that the "pooled results showed that there was a significant difference in local recurrence between the two radiotherapy techniques" was a result only for long-term, which is > 5 years of follow-up.
-The first part of the discussion should first summarize the meta results.
-In the introduction, please change "female worldwide" to "women worldwide"
-In the inclusion and exclusion criteria, how was it decided? For instance why was the cut-off set at studies who included less than 20 patients?
-In the results section, local recurrence, please change the font size to be the same as that used in the rest of the manuscript.
-There are multiple studies of the same patient population (TARGIT-A and 10 year follow-up) and ELIOT v.1 and follow up. The principles of the meta analysis are therefore not valid as there are duplicate patient cohorts/results analyzed.
(1)The Italian ELIOT RCT has been included twice (U Veronesi 2013 and R Orecchia 2021)
(2)TARGIT RCT has also been included thrice (Vaidya 2013, 2020 and 2021).
-Figures did not format correctly and are difficult to read.
-Table 1 format is unclear and needs to be adjusted.
-Abstract has several sentences that should be revised/edited for clarity if manuscript is to be resubmitted.
-Table 2 is Incomplete without data of DFS and OS.
-The entry represented by V1-7 shall be indicated in Table 3.

Reviewer 4 ·

Basic reporting

This paper used a meta-analysis to analyze the difference in efficacy of IORT and EBRT in the prevention of long-term LR in patients with early breast cancer, and to analyze the efficacy of the two methods in the prevention of short-term LR(within 5 years). The study has the following advantages: First, the authors included randomized controlled trials and high-quality subgroup studies with propensity matching scores. Second, short-term (5 years) and long-term (15 years) follow-up outcomes were evaluated. Finally, the authors' meta-analysis had a large sample size and participants covered multiple centers in different regions, largely avoiding possible racial and social influences. These are very meaningful, but there are some places in the paper that need to be improved, the specific suggestions are as follows:

1.It is noted that your manuscript needs careful editing by someone with expertise in technical English editing paying particular attention to English grammar, spelling, and sentence structure so that the goals and results of the study are clear to the reader.
2.Referencing style is inconsistent. In the text or table at some places the author’s first name and at other places their last name is mentioned along with the year of publication without citing the reference number. For some references, the journal names and extensions are not appropriate.
3.For reference 18 (Moini Nazi et al) the name of the journal is not mentioned.
4.Reference 20 (Preskorn SH,. Effect of Sublingual Dexmedetomidine vs Placebo on Acute Agitation Associated With Bipolar Disorder: A Randomized Clinical Trial. Jama. 2022;327(8):727-36) is clearly irrelevant to this article.
5.In the introduction part, the introduction structure of IORT, EBRT, IOXRT, IOERT, APBI is chaotic.

Experimental design

1.Please provide the retrieval records of 4 databases in the attachment.
2.It has included 3 retrospective studies (Kolberg 2017, Yin 2020, Moini 2020). In a meta-analysis of RCTs retrospective studies are not included.
3.Patients in the German study (Yasser 2019) included separately in the meta-analysis is part of the TARGIT-A RCT of Vaidya et al and therefore they are included twice in the meta-analysis.
4.The study by Antonella Ciabattoni 2021 is not comparing APBI with IORT with Whole Breast RT using EBRT but comparing the 2 methods for giving tumour bed boost after whole breast RT. This study cannot be combined with other RCTs comparing APBI (with IORT) versus Whole Breast RT (with external beam RT).
5.The principle of RT needs more definition in the article.
6.Differentiation between IOXRT and IOERT is not well difined in the article and needs more consideration.

Validity of the findings

no comment

Additional comments

In summary, I recommend the work should be acceptable after authors improve the above aspects, because the topic and model may be very attractive.

---

## Round 0.2 · accepted · Accept

In carefully evaluating the content of this revised paper, I was satisfied with the responses and revisions made by the authors. The Reviewer's and my concerns have been well addressed. With the necessary revisions and improvements, the quality of this paper has been significantly improved. I believe that this revised manuscript is ready to be considered for publication in PeerJ.

Reviewer 2 ·

Basic reporting

no comment

Experimental design

no comment

Validity of the findings

no comment

Additional comments

I appreciate the effort you've made for the revision, which is marked in different color in the manuscript. The paper is logical with a concise ordering of ideas and the quality of the readability of the paper in English is great. The ideas in the paper are original. My suggestion is the paper can be accepted for publication.

Reviewer 3 ·

Basic reporting

no comment

Experimental design

no comment

Validity of the findings

no comment

Additional comments

The paper has been revised and improved according to the reviewers' suggestions. The paper's structure is complete and the abstract summarizes the core idea of the paper. And the conclusion part and the usage of English are great too. All in all, this paper is qualified and I recommend it can be published in the journal.

Reviewer 4 ·

Basic reporting

The manuscript has been revised according to my comments, and I agree to accept.

Experimental design

The experimental design of this study is rigorous and comprehensive.

Validity of the findings

The results of this study have reliability and effectiveness.

Additional comments

None